# Individual long non-coding RNAs have no overt functions in zebrafish embryogenesis, viability and fertility

Mehdi Goudarzi[1]*, Kathryn Berg[1], Lindsey M Pieper[1], Alexander F Schier[1,2,3,4,5]*

[1]Department of Molecular and Cellular Biology, Harvard University, Cambridge, United States; [2]Center for Brain Science, Harvard University, Cambridge, United States; [3]FAS Center for Systems Biology, Harvard University, Cambridge, United States; [4]Allen Discovery Center for Cell Lineage Tracing, University of Washington, Seattle, United States; [5]Biozentrum, University of Basel, Basel, Switzerland

**Abstract** Hundreds of long non-coding RNAs (lncRNAs) have been identified as potential regulators of gene expression, but their functions remain largely unknown. To study the role of lncRNAs during vertebrate development, we selected 25 zebrafish lncRNAs based on their conservation, expression profile or proximity to developmental regulators, and used CRISPR-Cas9 to generate 32 deletion alleles. We observed altered transcription of neighboring genes in some mutants, but none of the lncRNAs were required for embryogenesis, viability or fertility. Even RNAs with previously proposed non-coding functions (*cyrano* and *squint*) and other conserved lncRNAs (*gas5* and *lnc-setd1ba)* were dispensable. In one case (*lnc-phox2bb*), absence of putative DNA regulatory-elements, but not of the lncRNA transcript itself, resulted in abnormal development. LncRNAs might have redundant, subtle, or context-dependent roles, but extrapolation from our results suggests that the majority of individual zebrafish lncRNAs have no overt roles in embryogenesis, viability and fertility.
DOI: https://doi.org/10.7554/eLife.40815.001

*For correspondence:
mgoudarzi@fas.harvard.edu (MG);
schier@fas.harvard.edu (AFS)

**Competing interests:** The authors declare that no competing interests exist.

## Introduction

Long non-coding RNAs (lncRNAs) comprise a heterogeneous group of transcripts longer than 200 nucleotides that do not encode proteins. LncRNAs have been proposed to affect the expression of neighboring or distant genes by acting as signaling, guiding, sequestering or scaffolding molecules (*St Laurent et al., 2015*; *Rinn and Chang, 2012*; *Nagalakshmi et al., 2008*; *Carninci et al., 2005*; *Kapranov et al., 2007*). The functions of specific lcnRNAs in dosage compensation (*xist* (*Brockdorff et al., 1991*; *Marahrens et al., 1997*), *tsix* (*Lee et al., 1999*), *jpx* (*Johnston et al., 2002*)) and imprinting (*Airn* (*Wutz et al., 1997*; *Latos et al., 2012*), *MEG3* (*Miyoshi et al., 2000*; *Kobayashi et al., 2000*), *H19* (*Bartolomei et al., 1991*; *Feil et al., 1994*)) are well established, and mutant studies in mouse have suggested that *fendrr, peril, mdget, linc-brn1b, linc-pint* (*Sauvageau et al., 2013*), and *upperhand* (*Anderson et al., 2016*) are essential for normal development. However, other studies have questioned the developmental relevance of several mouse lncRNAs, including *Hotair* (*Amândio et al., 2016*), *MIAT/Gumafu* (*Ip et al., 2016*), *Evx1-as* (*Bell et al., 2016*), *upperhand, braveheart* and *haunt* (*Han et al., 2018*). In zebrafish, morpholinos targeting the evolutionarily conserved lncRNAs *megamind* (TUNA (*Lin et al., 2014*)) and *cyrano* resulted in embryonic defects (*Ulitsky et al., 2011*). However, a mutant study found no function for *megamind* and revealed that a *megamind* morpholino induced non-specific defects (*Kok et al., 2015*). These conflicting results have led to a controversy about the importance of lncRNAs for vertebrate development (*Sauvageau et al., 2013*), (*Han et al., 2018*). We therefore decided to mutate

a group of selected zebrafish lncRNAs using CRISPR-Cas9, and assay their roles in embryogenesis, viability and fertility.

Transcriptomic studies of early embryonic development (*Ulitsky et al., 2011*; *Pauli et al., 2012*) and five adult tissues (*Kaushik et al., 2013*) have identified over 2000 lncRNAs in zebrafish (*Dhiman et al., 2015*), of which 727 have been confirmed as non-coding based on ribosome occupancy patterns (*Chew et al., 2013*). For our mutant analysis we selected 24 bona fide lncRNAs based on synteny (conserved relative position on at least one other vertebrate genome), sequence conservation, expression dynamics (expression levels, onset and pattern) and proximity to developmental regulatory genes (see *Table 1*). These criteria were chosen to increase the likelihood of potential functional requirements of the selected lncRNAs. In addition, we selected a protein-coding RNA with a proposed non-coding function (*squint*).

## Results and discussion

The genomic location of selected lncRNAs are depicted in *Figure 1*. The neighbor-relationship, and expression levels of the selected lncRNAs and their neighboring genes are shown in *Figure 1—figure supplement 1–1*, *Figure 1—figure supplement 1–2*, respectively.

Using CRISPR-Cas9 (*Figure 1—figure supplement 1–3*) we generated 32 knockout-alleles. 24 alleles removed regions containing transcription start sites (TSS-deletion; 244 bp to 736 bp), and eight alleles fully or partially removed the gene (1 kb to 203 kb) (*Table 1*). qRT-PCR analysis demonstrated effective reduction in the levels of the targeted lncRNA transcripts (average reduction of 94 ± 6%; *Table 1*), which was further tested and confirmed for a subset of lncRNAs by in situ RNA hybridization (*Figures 2B*, *3B, C*, *4D*, *5B* and *6D*).

Previous observations in mammalian cell culture systems suggested that lncRNA promoters can affect the expression of nearby genes (*Engreitz et al., 2016*). To test if these results hold true in vivo, we measured the changes in the expression of neighboring genes (a 200 kb window centered on each lncRNA) in lncRNA mutants. Several mutants displayed changes in the expression of neighboring genes (*Figure 1—figure supplement 1–4*). In particular, 10 out of 40 neighboring genes showed more than two-fold changes in expression, lending in vivo support to observations in cell culture systems (*Engreitz et al., 2016*).

To determine the developmental roles of our selected lncRNAs, we generated maternal-zygotic mutant embryos (lacking both maternal and zygotic lncRNA activity) and analyzed morphology from gastrulation to larval stages, when all major organs have formed. Previous large-scale screens (*Driever et al., 1996*; *Haffter et al., 1996*) have shown that the visual assessment of live embryos and larvae is a powerful and efficient approach to identify mutant phenotypes, ranging from gastrulation movements and axis formation to the formation of brain, spinal cord, floor plate, notochord, somites, eyes, ears, heart, blood, pigmentation, vessels, kidney, pharyngeal arches, head skeleton, liver, and gut. No notable abnormalities were detected in 31/32 mutants. Moreover, these 31 mutants survived to adulthood, indicating functional organ physiology, and were fertile (*Table 1*). In the following section, we describe the results for five specific lncRNAs and put them in the context of previous studies.

### Cyrano

*cyrano* is evolutionarily conserved lncRNA and based on morpholino studies, has been suggested to have essential functions during zebrafish embryogenesis (*Ulitsky et al., 2011*) and brain morphogenesis (*Sarangdhar et al., 2018*). *cyrano* has also been suggested to act as a sponge (decoy-factor) for HuR during neuronal proliferation (*Kim et al., 2016a*), regulate *miR-7* mediated embryonic stem cell differentiation (*Smith et al., 2017*), and control the level of *miR-7* in the adult mouse brain (*Kleaveland et al., 2018*). We generated two mutant alleles that removed the TSS (*cyrano*[a171]) or the gene (*cyrano*[a172]), including the highly conserved *miR-7* binding-site (*Figure 2A,B*). The expression level of the nearby gene (*oip5*) was not affected in either of these mutants (*Figure 1—figure supplement 1–4*). In contrast to previous morpholino studies in zebrafish (*Ulitsky et al., 2011*) but in support of recent findings in mouse (*Kleaveland et al., 2018*), *cyrano* mutants developed normally and were viable and fertile.

The difference between morphant (*Ulitsky et al., 2011*) and mutant phenotypes might be caused by compensation in the mutants (*Rossi et al., 2015*; *El-Brolosy and Stainier, 2017*). To test this

**Table 1.** Summary of lncRNA features and mutant phenotypes lncRNA names are shown in the first column.
lncRNAs were named using the last four digits of their corresponding ENSEMBL Transcript ID or their chromosome number if no transcript ID was available (e.g. lnc-1200 is located on chromosome 12). The second column represents ribosomal occupancy pattern along the length of lncRNAs in comparison to the 5'UTR, coding and 3'UTR of typical protein-coding transcripts (*Chew et al., 2013*). The third column shows the transcript ID for the investigated lncRNA or its genomic coordinate in GRCz10. Column Four shows the deletion size. Fifth column represent the percentage decrease in the level of lncRNA in comparison to wild type from three biological replicates (qRT-PCR). The six and seven columns show the presence of embryonic phenotypes, viability and fertility (at least 15 adult pairs per allele) of homozygous mutant fish. Eighth and ninth column show the upstream and downstream neighboring genes in a 200 kb window centered around the lncRNA's TSS. The last column provides the selection criteria for each lncRNA.

| lncRNA mutant, deletion type | Ribosome Profiling, class | lncRNA transcript ID | Deletion size | Percent reduction | Embryonic phenotype | Viability and fertility | Neighboring genes Up 100 Kb | Down 100 Kb | Selection criteria |
|---|---|---|---|---|---|---|---|---|---|
| cyrano[a171], TSS-del. | Trailerlike | ENSDART0 0000139872 | 326 bp | 98% | No | Yes | tmem39b | oip5 | Syntenic and sequence conservation, Reported phenotype |
| cyrano[a172], gene del. | Trailerlike | ENSDART0 0000139872 | 4374 bp | 94% | No | Yes | tmem39b | oip5 | Syntenic and sequence conservation, Reported phenotype |
| gas5[a173], TSS-del. | Leaderlike | ENSDART0 0000156268 | 296 bp | 100% | No | Yes | osbpl9 | tor3a | Syntenic conservation, well studied lncRNA, host of several snoRNA |
| lnc-setd1ba[a174], gene del. | Leaderlike | ENSDART0 0000141500 | 3137 bp | 100% | No | Yes | setd1ba | rhoF | Syntenic and sequence conservation, Proximity to developmental regulatory genes |
| squint[a175], gene del. | Coding | ENSDART0 0000079692 | 1032 bp | 95% | No | Yes | htr1ab | eif4ebp1 | Evolutionary conservation, Reported phenotype, putative cncRNA |
| lnc-phox2bb[a176], TSS-del. | Leaderlike | ENSDART0 0000158002 | 652 bp | 99% | No | Yes | smntl1 | phox2bb | Syntenic conservation |
| lnc-phox2bb[a177], gene del. | Leaderlike | ENSDART00 000158002 | 9361 bp | 87% | Yes | No | smntl1 | phox2bb | Syntenic conservation |
| lnc-3852[a178], TSS-del. | Leaderlike | ENSDART00 000153852 | 447 bp | 100% | No | Yes | lima1a | hoxc1a | Maternal expression, Proximity to developmental regulatory genes |
| lnc-1562[a179], TSS-del. | Leaderlike | ENSDART00 000131562 | 409 bp | 90% | No | Yes | * | fgf10a | Maternal expression, Proximity to developmental regulatory genes |
| lnc-3982[a180], TSS-del. | Leaderlike | ENSDART00 000153982 | 352 bp | 97% | No | Yes | * | bmp2b | Maternal expression, Proximity to developmental regulatory genes |
| lnc-6269[a181], TSS-del. | Leaderlike | ENSDART00 000156269 | 535 bp | 99% | No | Yes | tbx1 | * | Maternal expression, Proximity to developmental regulatory genes |
| lnc-2154[a182], TSS-del. | Trailerlike | ENSDART00 000132154 | 546 bp | 100% | No | Yes | rpz | nr2f5 | Maternal expression, Proximity to developmental regulatory genes |
| lnc-1200[a183], TSS-del. | Leaderlike | Chr12:1708389-1925779:1 | 590 bp | 95% | No | Yes | * | zip11 | Maternal expression, Longest selected lncRNA |
| lnc-1200[a184], gene del. | Leaderlike | Chr12:1708389-1925779:1 | 203.8 kb | 84% | No | Yes | * | zip11 | Maternal expression, Longest selected lncRNA |
| lnc-2646[a185], TSS-del. | Leaderlike | ENSDART00 000152646 | 240 bp | 97% | No | Yes | * | dkk1b | Proximity to developmental regulatory genes |
| lnc-4468[a186], TSS-del. | Leaderlike | ENSDART00 000154468 | 306 bp | 100% | No | Yes | fam169ab | lhx5 | Proximity to developmental regulatory genes, Low expression level |

*Table 1 continued on next page*

*Table 1 continued*

| lncRNA mutant, deletion type | Ribosome Profiling, class | lncRNA transcript ID | Deletion size | Percent reduction | Embryonic phenotype | Viability and fertility | Neighboring genes Up 100 Kb | Down 100 Kb | Selection criteria |
|---|---|---|---|---|---|---|---|---|---|
| *lnc-0600*[a187], TSS-del. | Trailerlike | Chr6:59414652-59443141:1 | 244 bp | 95% | No | Yes | * | *gli1* | Proximity to developmental regulatory genes, Low expression level |
| *lnc-0900*[a188], TSS-del. | Leaderlike | Chr9:6684669-6691350:1 | 377 bp | 83% | No | Yes | *pou3f3a* | * | Syntenic conservation, Low expression level |
| *lnc-8507*[a189], mTSS-del. | Leaderlike | ENSDART00000158507 | 323 bp | 81% | No | Yes | *npvf* | *hoxa1a* | Proximity to Hox genes, Maternal and Zygotic promoters |
| *lnc-8507*[a190], mzTSS-del. | Leaderlike | ENSDART00000158507 | 9773 bp | 95% | No | Yes | *npvf* | *hoxa1a* | Proximity to Hox genes, Maternal and Zygotic promoters |
| *lnc-7620*[a191], TSS-del. | Trailerlike | ENSDART00000137620 | 668 bp | 99% | No | Yes | *gal3st1b* | *srsf9* | Syntenic and sequence conservation, Implicated in adult fish and mouse behavior. Bitetti, A., et al. (2018) |
| *lnc-1300*[a192], TSS-del. | Leaderlike | Chr13:4535992-4538275:1 | 367 bp | 92% | No | Yes | *c1d* | *pla2g12b* | Syntenic and sequence conservation, High expression level |
| *lnc-7118*[a193], TSS-del. | Trailerlike | ENSDART0000157118 | 438 bp | 82% | No | Yes | *mrps9* | *pou3f3b* | Syntenic conservation |
| *lnc-5888*[a194], TSS-del. | Leaderlike | ENSDART00000155888 | 606 bp | 96% | No | Yes | *glrx5* | *zgc:100997* | Syntenic conservation, scaRNA13 host gene, shortest selected lncRNA |
| *lnc-6913*[a195], TSS-del. | Leaderlike | ENSDART00000156913 | 333 bp | 72% | No | Yes | *usp20* | *ptges* | Proximity to developmental regulatory genes |
| *lnc-6913*[a196], gene del. | Leaderlike | ENSDART00000156913 | 5568 bp | 93% | No | Yes | *usp20* | *ptges* | Proximity to developmental regulatory genes |
| *lnc-1666*[a197], TSS-del. | Leaderlike | ENSDART00000141666 | 544 bp | 96% | No | Yes | *ptf1a* | * | Proximity to developmental regulatory genes, Restricted late expression |
| *lnc-6490*[a198], TSS-del. | Leaderlike | ENSDART0000146490 | 607 bp | 99% | No | Yes | *nr2f2* | * | Syntenic conservation, Restricted late expression |
| *lnc-6490*[a199], gene del. | Leaderlike | ENSDART0000146490 | 8378 bp | 100% | No | Yes | *nr2f2* | * | Syntenic conservation, Restricted late expression |
| *lnc-0464*[a200], TSS-del. | Trailerlike | ENSDART00000140464 | 597 bp | 96% | No | Yes | *nr2f1a* | * | Restricted late expression pattern |
| *lnc-4149*[a201], TSS-del. | Leaderlike | ENSDART00000154149 | 491 bp | 98% | No | Yes | *bhlhe22* | * | Proximity to developmental regulatory genes |
| *lnc-4149*[a202], gene del. | Leaderlike | ENSDART00000154149 | 35.11 kb | 100% | No | Yes | *bhlhe22* | * | Proximity to developmental regulatory genes |

DOI: https://doi.org/10.7554/eLife.40815.002

possibility, we injected the previously used morpholinos targeting the first exon-intron boundary (e1i1) or the conserved *miR-7* binding site (CMiBS) into wild type and homozygous deletion mutants. The TSS-mutant allele lacked the e1i1 morpholino-binding site and the gene deletion allele lacked the CMiBS morpholino-binding site (*Figure 2A*). The previously reported phenotypes, including

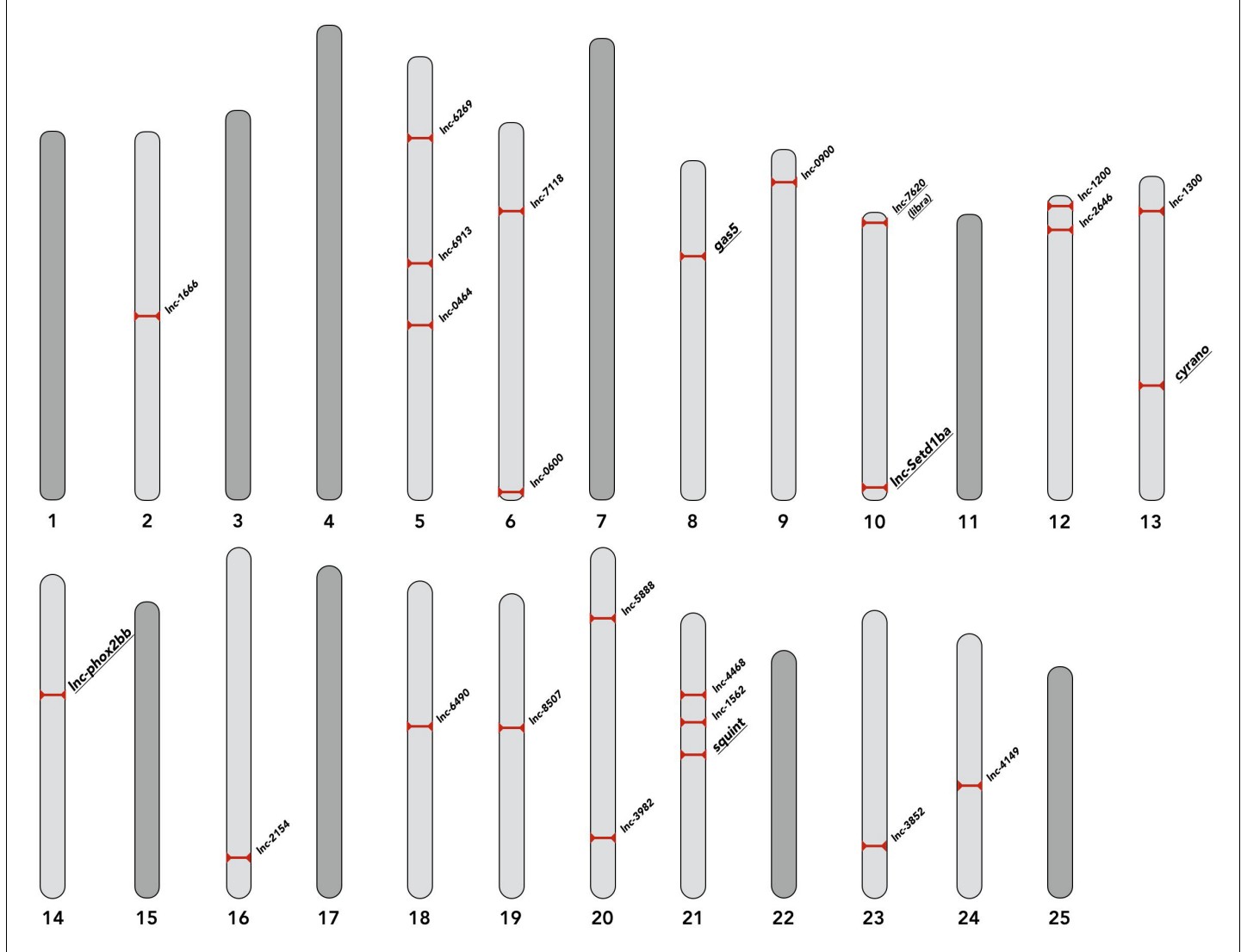

**Figure 1.** Genomic location of selected lncRNAs. The chromosomal positions of selected lncRNAs are depicted. lncRNAs discussed in the text are underlined. The corresponding genomic coordinates for all lncRNAs are provided in the **supplementary file 2**.

DOI: https://doi.org/10.7554/eLife.40815.003

The following figure supplements are available for figure 1:

**Figure supplement 1.** Size, relative distance and orientation of selected lncRNAs and their neighboring genes

DOI: https://doi.org/10.7554/eLife.40815.004

**Figure supplement 2.** Expression levels of selected lncRNAs and their neighboring protein-coding genes.

DOI: https://doi.org/10.7554/eLife.40815.005

**Figure supplement 3.** Cas9-mediated deletion approach for generating lncRNA knockouts 6 gRNAs (three at either side of the TSS) were used to remove TSS.

DOI: https://doi.org/10.7554/eLife.40815.006

**Figure supplement 4.** Summary of qRT-PCR analysis for lncRNA and their neighboring genes.

DOI: https://doi.org/10.7554/eLife.40815.007

small heads and eyes, heart edema, and kinked tails were found in both wild type and mutants (**Figure 2C**), demonstrating that the morpholino-induced phenotypes were non-specific. These results reveal that *cyrano* transcripts or their evolutionarily conserved *miR-7*-binding site, are not required for embryogenesis, viability or fertility.

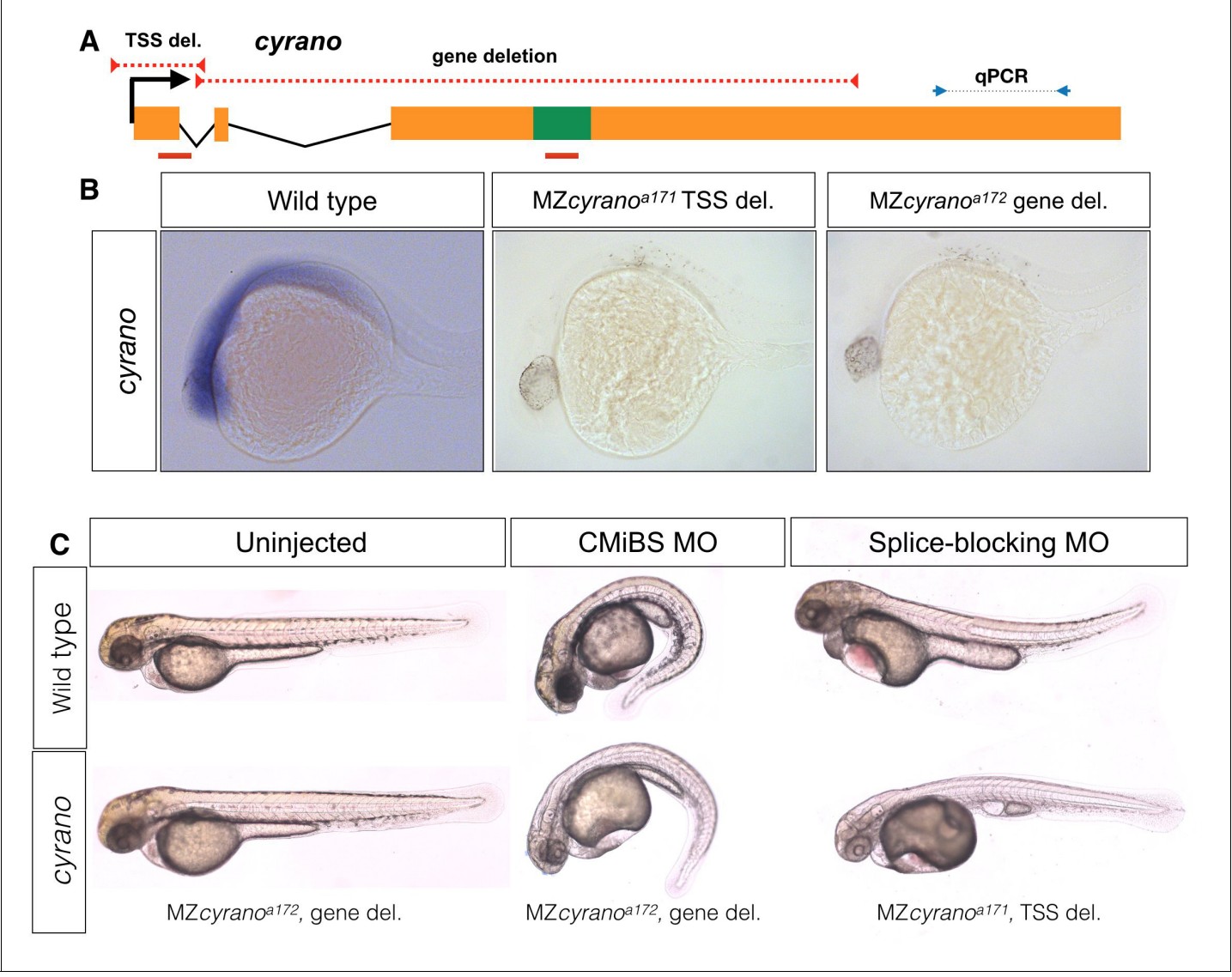

**Figure 2.** Normal embryogenesis of *cyrano* mutants. (**A**) The positions of TSS-deletion allele and gene deletion allele are marked by dashed red lines. Green box represents the conserved element in *cyrano* which is complementary to *miR-7*. Solid red lines indicate the position of the first exon-intron boundary (e1i1) morpholino and conserved microRNA binding site (CMiBS) morpholinos. Arrows flanking black dotted line mark the primer binding sites for qRT-PCR product. (**B**) Representative images of in situ hybridization for *cyrano* in wild type (15/15) and both homozygous TSS-deletion (21/22) and gene deletion (18/18) 1-dpf. (**C**) At 2-dpf gene deletion mutants (lower-left), (and TSS-deletion mutants, not shown) were not different from the wild-type embryos (upper-left). Morpholino injected wild-type embryos (upper-middle and upper-left) reproduced observed phenotype in Ulitsky et. al (*Kok et al., 2015*). Morpholino injected deletion-mutants, lacking the corresponding binding sites for morpholinos, (lower-middle and lower-left) were comparable to morpholino injected wild types.

DOI: https://doi.org/10.7554/eLife.40815.008

## gas5

*gas5* is an evolutionarily conserved lncRNA (*growth-arrest specific 5*) (*Coccia et al., 1992*) that is highly expressed in early development (*Figure 3B*) and hosts several snoRNAs implicated in zebra-fish development (*Higa-Nakamine et al., 2012*). Knockdown and knockout studies in cell culture (*Ma et al., 2016*) have indicated that *gas5* might act as a tumor suppressor (*Pickard and Williams, 2015*) and exert effects at distant genomic sites (*Schneider et al., 1988*). However, the role of this lncRNA in development has not been studied in any vertebrate. Our *gas5^a173* mutant allele removed the sequences containing the TSS (−169 to +127) (*Figure 3A*) and resulted in complete elimination of its expression (*Figure 3B and D*). Expression of the neighboring gene *osbpl9*, encoding a lipid-

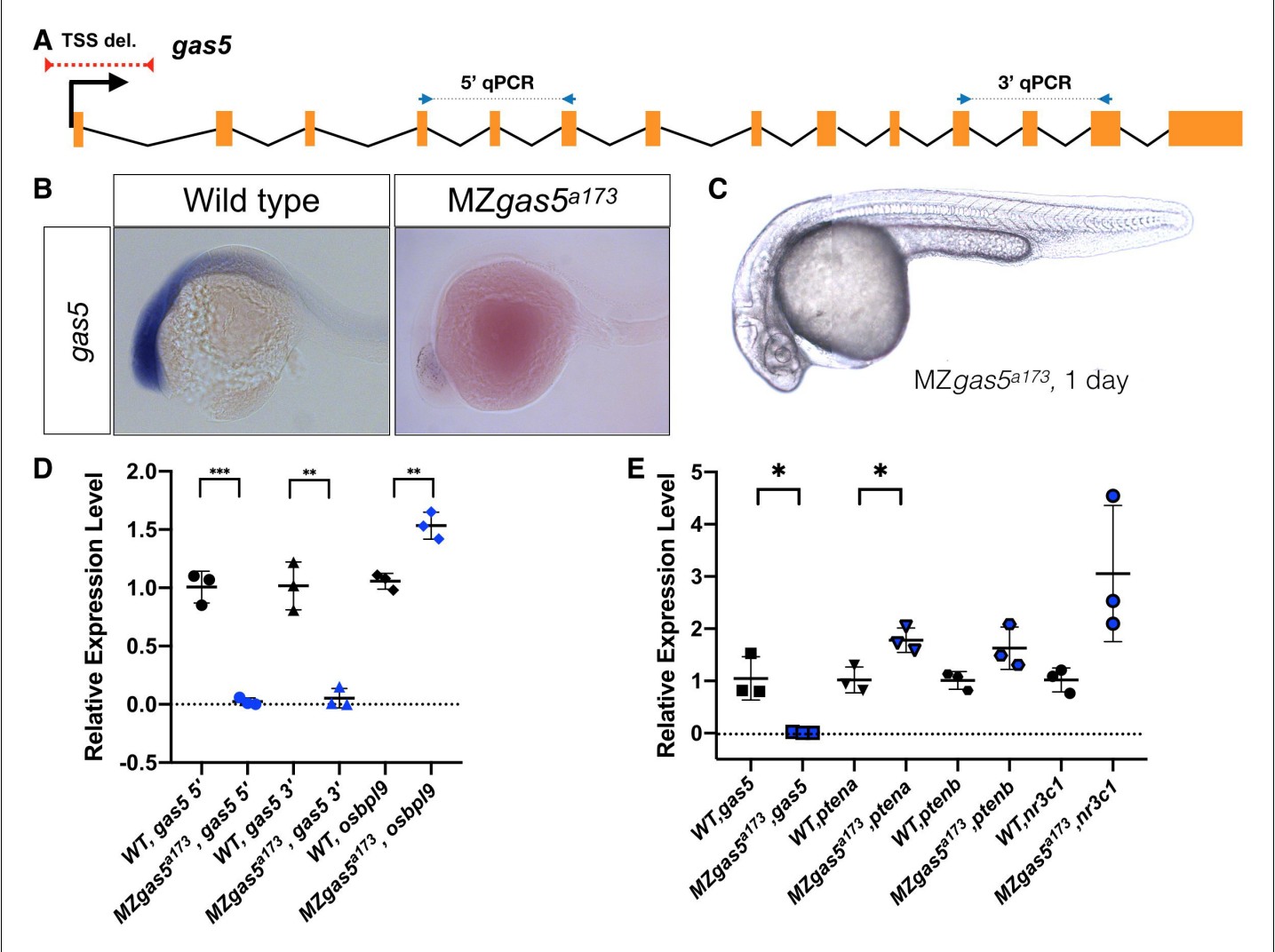

**Figure 3.** Normal embryogenesis of *gas5* mutants. (A) Position of the TSS-deletion allele in *gas5* is marked by dashed red line. Arrows flanking black dotted lines mark the primer binding sites for 5'-qPCR and 3'-qPCR products. (B) Representative in situ hybridization images for *gas5* in wild type (11/11) and homozygous TSS-deletion mutants (11/11). (C) Maternal and Zygotic gas5 (MZ*gas5*) mutant embryos at 1-dpf were indistinguishable from the wild-type embryos at the same developmental stage (not shown). (D) Expression level of *gas5* and osbpl9 measured by qRT-PCR. Tor3A, the other neighboring gene, was not expressed at the investigated time-point. (E) Expression level of *gas5,* its trans targets *ptena*, *ptenb* and *nr3c1* measured by qRT-PCR. The statistical significance of the observed changes was determined using t-test analysis and represented by star marks (*, **, ***, and **** respectively mark p-values<0.05,<0.01,<0.001 and<0.0001).

DOI: https://doi.org/10.7554/eLife.40815.009

binding protein, was increased by 50% (*Figure 3D*). Previous studies have shown that *gas5* lncRNA can act in trans to affect *pten* expression (*ptena* and *ptenb* in zebrafish) by sequestering specific microRNAs (*Li et al., 2017*; *Zhang et al., 2018*; *Liu et al., 2018*). Additionally, *gas5* transcript can mimic Glucocorticoid Response Element and act as a decoy factor (riborepressor) for the Glucocorticoid Receptor (nr3c1)-mediated transcription (*Kino et al., 2010*). We analyzed the expression level changes of these genes in MZgas5[a173] embryos (at 1-dpf) and found significant upregulation for *ptena* in MZgas5[a173] mutants (*Figure 3E*). Despite these changes in gene expression, *gas5[a173]* mutants were indistinguishable from wild type (*Figure 3C*), reached adulthood and were fertile.

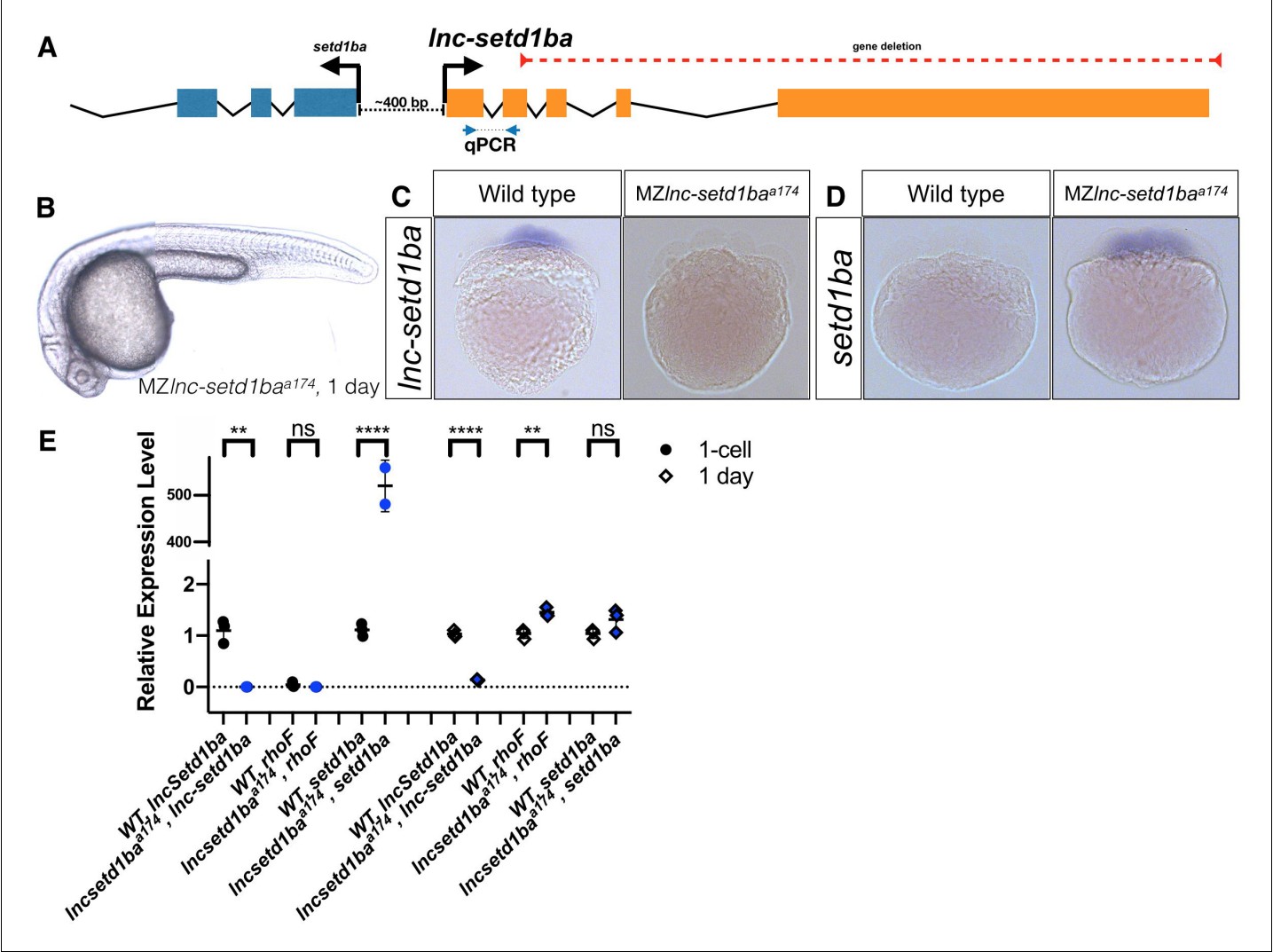

**Figure 4.** Normal embryogenesis of *lnc-setd1ba* mutants. (**A**) The relative position of *lnc-setd1ba* and the protein-coding gene *setd1ba*. The gene deletion region is marked by dashed red line. Arrows flanking black dotted line mark the primer-binding sites for qRT-PCR product. (**B**) Maternal and zygotic *lnc-setd1ba* mutants were not different from wild-type embryos at 1-dpf. (**C**) Representative images of in situ hybridization for *lnc-setd1ba* at four- to eight-cell stage mutant (18/18) and wild-type (25/25) embryos. (**D**) In situ hybridization for the protein-coding mRNA, *setd1ba* (9/11) in *lnc-setd1ba* mutants compared to the wild-type embryos (15/15). (**E**) qRT-PCR at 1 cell stage and 1-dpf for the lncRNA and its neighboring genes *rhoF* and *setd1ba*. The statistical significance of the observed changes was determined using t-test analysis and represented by star marks (ns, *, **, ***, and **** respectively mark p-values≥0.05,<0.05,<0.01,<0.001 and<0.0001).

DOI: https://doi.org/10.7554/eLife.40815.010

## Lnc-setd1ba

*Lnc-setd1ba* is the zebrafish orthologue of human LIMT (**Sas-Chen et al., 2016**) (LncRNA Inhibiting Metastasis), which has been implicated in basal-like breast cancers. It is expressed from a shared promoter region that also drives the expression of the histone methyltransferase *setd1ba* in opposite direction (**Figure 4A**). Evolutionary conservation in vertebrates and proximity to *setd1ba*, whose mouse homolog is essential for embryonic development (**Eymery et al., 2016**; **Kim et al., 2016b**) prompted us to investigate the function of this lncRNA in zebrafish. We removed the gene of *lnc-setd1ba* downstream of its TSS (3137 bp deletion) (*lnc-setd1ba*[a174]). In situ hybridization and qRT-PCR revealed absence of lncRNA expression (**Figure 4C and E**) and strong upregulation of *setd1ba* (**Figure 4D and E**) during cleavage stages and slight upregulation of *setd1ba* and the other neighboring gene *rhoF* at one-day post fertilization (1-dpf) (**Figure 4E**). Despite these changes, maternal-

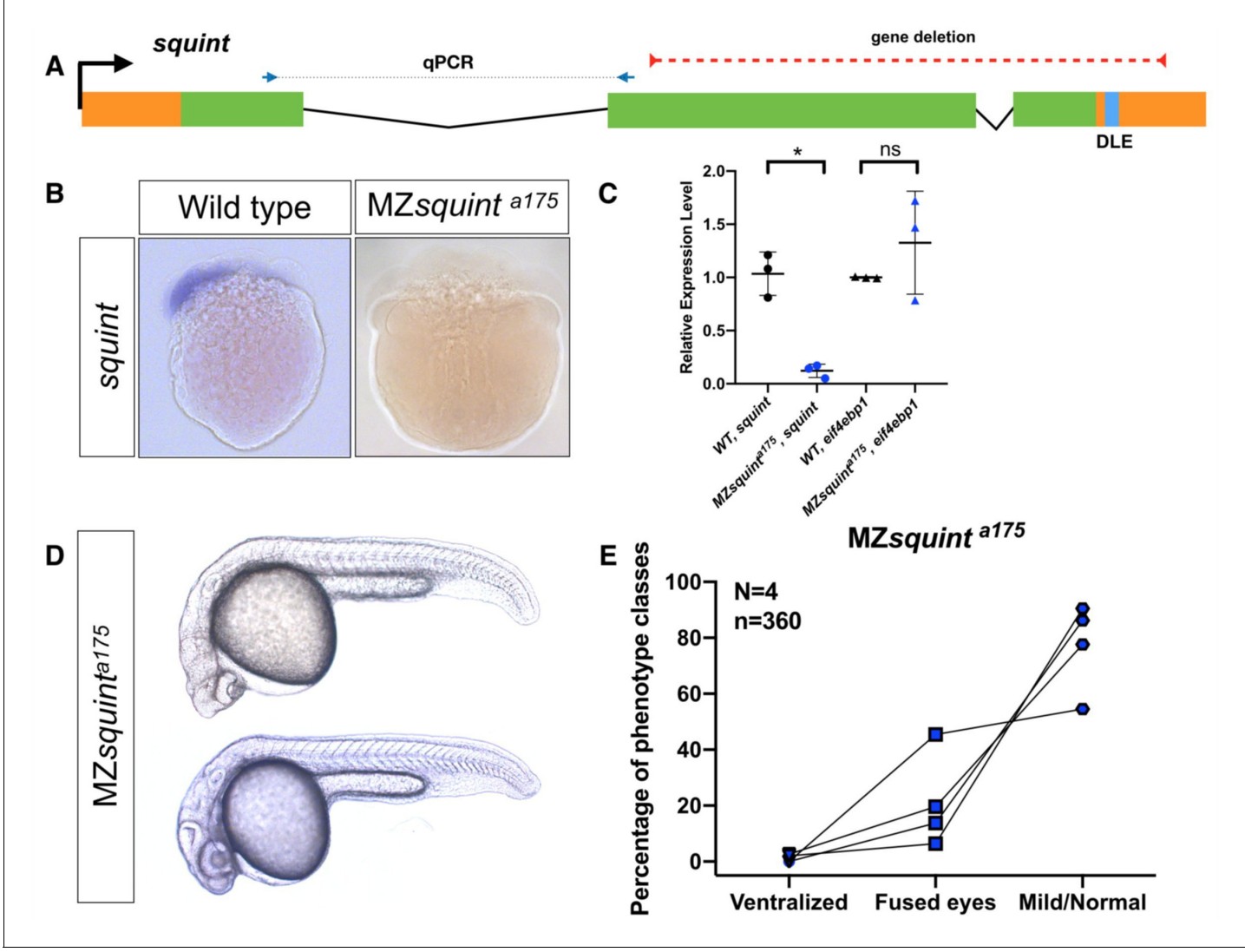

**Figure 5.** No non-coding function for *squint* 3'UTR. (**A**) The position of untranslated regions (brown), coding region (green), putative Dorsal Localization Element- DLE (blue) and the gene deletion (red dashed line) in the *squint* genomic locus. Arrows flanking black dotted line mark the primer binding sites for qRT-PCR product. (**B**) In situ hybridization for *squint* at 8-cell stage on wild-type (18/20) and MZ*squint*^a175^(17/17) embryos. (**C**) qRT-PCR for *squint* and *eif4ebp1* on wild-type and MZ*squint*^a175^ embryos at 1-cell stage. (**D**) Two representative MZ*squint*^a175^ embryos. (**E**) MZ*squint*^a175^ embryonic phenotype (N = 4 independent crosses, n = 360 embryos). The statistical significance of the observed changes was determined using t-test analysis and represented by star marks (ns, *, **, ***, and **** respectively mark p-values≥0.05,<0.05,<0.01,<0.001 and<0.0001).
DOI: https://doi.org/10.7554/eLife.40815.011

The following figure supplement is available for figure 5:

**Figure supplement 1.** Dorsalization induced by Overexpression of *squint* mRNA but not its non-protein coding version.
DOI: https://doi.org/10.7554/eLife.40815.012

zygotic *lnc-setd1ba*^a174^ mutants were indistinguishable from wild type (*Figure 4B*), reached adulthood and produced normal progeny.

## Squint

*Squint* encodes a Nodal ligand involved in mesendoderm specification (*Pei et al., 2007*; *Heisenberg and Nüsslein-Volhard, 1997*). The previously studied *squint* insertion mutant alleles (*squint*^Hi975Tg 50^ and *squint*^cz35 51^) lead to delayed mesendoderm specification and partially penetrant cyclopia (*Dougan et al., 2003*). Morpholino and misexpression studies have suggested an additional, non-coding role for maternally provided s*quint*, wherein the *squint* 3'UTR mediates dorsal

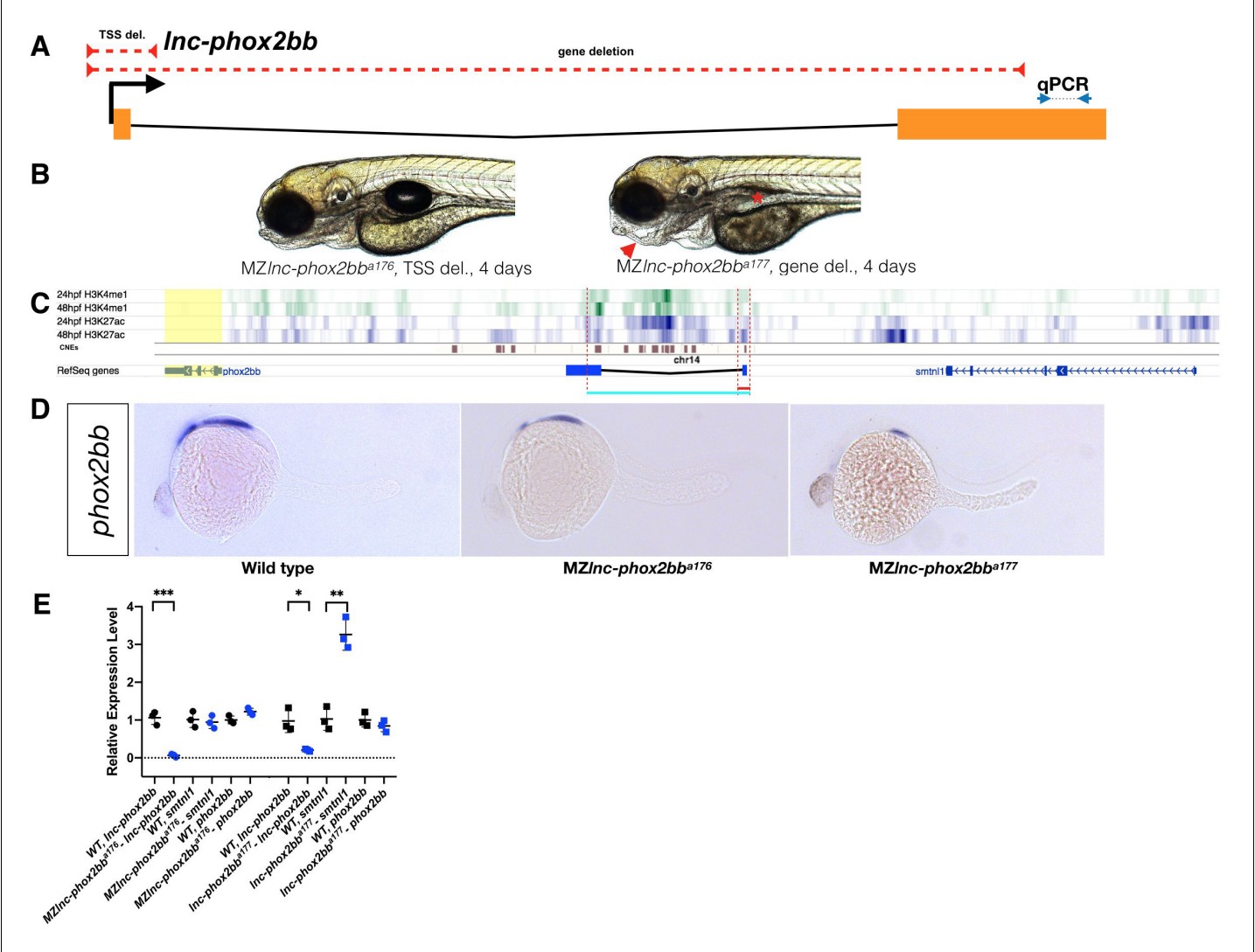

**Figure 6.** Requirement for *lnc-phox2bb* genomic elements but not RNA. (**A**) The red dashed lines depict the respective positions of the *lnc-phox2bb* TSS and gene deletion. Arrows flanking black dotted line mark the primer binding sites for qRT-PCR product. (**B**) Homozygous gene deletion mutants but not the TSS-deletion mutants show embryonic defects in jaw formation (arrow head) and swim bladder inflation (asterisk) by 4-dpf. (**C**) Histone marks (H3K4me1 and H3K27ac) associated with enhancer activity (**Bogdanovic et al., 2012**) and conserved noncoding elements (CNEs) (**Hiller et al., 2013**) overlap with gene deletion. (**D**) phox2bb expression pattern in the TSS and gene deletions. (**E**) qRT-PCR analysis on MZ TSS-deletion and gene deletion mutants. The statistical significance of the observed changes was determined using t-test analysis and represented by star marks (*, **, ***, and **** respectively mark p-values<0.05,<0.01,<0.001 and<0.0001).
DOI: https://doi.org/10.7554/eLife.40815.013

localization of *squint* mRNA, induces the expression of dorsal mesoderm genes, and is required for the development of dorsal structures (**Gore et al., 2005**; **Lim et al., 2012**). This mode of activity assigns *squint* to the cncRNA family - RNAs with both protein-coding and non-coding roles (**Sampath and Ephrussi, 2016**). To investigate the non-coding roles of *squint* mRNA we generated a deletion allele (*squint*[a175]) that lacked most of the protein coding region and the 3'UTR, including the Dorsal Localization Element (DLE) implicated in maternal *squint* RNA localization (**Gilligan et al., 2011**) (**Figure 5A**). In this allele 525 bp (178 bp 5'UTR, 280 bp first exon and 67 bp of second exon) out of the 1592bp-long mature transcript remain in the genome (**Figure 5A**). In situ hybridization (**Figure 5B**) and qRT-PCR (**Figure 5C**) showed that the level of remaining *squint* transcript was greatly reduced (~90%). MZ*squint*[a175] embryos displayed partially penetrant cyclopia, similar to existing protein-disrupting *squint* alleles (**Figure 5D**) (**Pei et al., 2007**; **Heisenberg and Nüsslein-**

*Volhard, 1997*; *Golling et al., 2002*), but the defects proposed to be caused by interference with *squint* non-coding activity (*Gore et al., 2005*) were not detected.

To further test whether *squint* mRNA might have non-coding roles, we injected wild-type and MZ*squint* [a175] embryos with either control RNA, full-length *squint* mRNA, a non-coding version of *squint* mRNA, or the putative transcript produced in *squint* [a175] (*Figure 5—figure supplement 5–S1*). We found that in contrast to wild-type *squint* mRNA, control RNA, non-protein coding *squint* RNA or *squint* [a175] RNA did not cause any phenotypes and did not rescue MZ*squint* [a175] mutants. These results indicate that *squint 3'UTR* does not have the previously proposed non-coding functions and that the *squint* transcript may not be a member of the cncRNA family.

## Transcript-independent phenotype at *lnc-phox2bb* locus

*Lnc-phox2bb* neighbors *phox2bb* and *smtnl1*. Phox2bb is a transcription factor implicated in the development of the sympathetic nervous system (*Pei et al., 2013*), (*Moreira et al., 2016*; *Tolbert et al., 2017*), while smtnl1 has been implicated in smooth muscle contraction (*Borman et al., 2009*). Whole-gene deletion of *lnc-phox2bb* (*lnc-phox2bb[a177]*) (*Figure 6A*) led to jaw deformation and failure to inflate the swim-bladder (*Figure 6B*), and no homozygous mutant fish survived to adulthood. Like the whole-gene deletion allele, the TSS-deletion allele (*lnc-phox2bb[a176]*) lacked *lnc-phox2bb* RNA (*Figure 6E*), but in contrast to the whole-gene deletion mutants, TSS-deletion mutants developed normally and gave rise to fertile adults. To determine the cause of this difference, we analyzed the expression level and pattern of neighboring genes. We found that the anterior expression domain of *phox2bb* in the hindbrain was absent in the whole-gene deletion allele (*Figure 6D*). This finding is consistent with the observation that the deleted region contains enhancer elements for *phox2bb* (*McGaughey et al., 2008*), conserved non-coding elements (CNEs) (*Hiller et al., 2013*) (*Figure 6C*), and histone marks related to enhancer regions (H3K4me1 and H3K27Ac) (*Bogdanovic et al., 2012*). We also found that the expression level of *smtnl1* increased in gene deletion mutants relative to the TSS-deletion mutant and wild type (*Figure 6E*). These results indicate that *lnc-phox2bb* RNA is not required for normal development but that the *lnc-phox2bb* overlaps with regulatory elements required for proper expression of *phox2bb* and *smtnl1* (*Figure 6E*).

In summary, our systematic mutant studies indicate that none of the 25 lncRNAs analyzed here are essential for embryogenesis, viability or fertility, including the prominent lncRNAs *cyrano*, *gas5*, and *lnc-setd1ba*. Additionally, they refute the proposed non-coding function of *squint* RNA. Our phenotypic screen does not exclude more subtle phenotypes; for example in behavior or brain activity (*Rihel et al., 2010*; *Randlett et al., 2015*; *Summer et al., 2018*). This mutant collection can now be analyzed for subtle, context specific or redundant functions, but extrapolation suggests that most individual zebrafish lncRNAs are not required for embryogenesis, viability or fertility.

# Materials and methods

## Animal care

TL/AB zebrafish (Danio rerio) were used as wild-type fish in this study. Fish were maintained on daily 14 hr (light): 10 hr (dark) cycle at 28°C. All animal works were performed at the facilities of Harvard University, Faculty of Arts and Sciences (HU/FAS). This study was approved by the Harvard University/Faculty of Arts and Sciences Standing Committee on the Use of Animals in Research and Teaching (IACUC; Protocol #25–08)

## Cas9 mediated mutagenesis

Guide RNAs (gRNAs) were designed using CHOPCHOP (*Montague et al., 2014*) and synthesized in pool for each candidate as previously described (*Gagnon et al., 2014*). (See *supplementary file 1* for the gRNA sequences). gRNAs were combined with Cas9 protein (50 μM) and co-injected (~1 nL) into the one-cell stage TL/AB wild-type embryos. Genomic DNA from 10 injected and 10 un-injected siblings was extracted (*Meeker et al., 2007*) and screened for the difference in amplified band pattern from the targeted region (See *supplementary file 1* for the genotyping primer sequences). The rest of injected embryos were raised to adulthood, crossed to wild-type fish and screened for passing the mutant allele to the next generation. Founder fish with desirable mutations were selected

and confirmed by Sanger sequencing of the amplified mutant allele. Heterozygous mutants were crossed together to generate homozygous mutants. At least 15 adult homozygous mutant pairs per allele were crossed to test fertility of mutants and to generate maternal and zygotic mutants (MZ) devoid of maternally and zygotic lncRNA activity.

### Phenotype scoring procedure

Visual assessment of live embryos and larvae performed (*Driever et al., 1996*; *Haffter et al., 1996*) to identify mutant phenotypes, ranging from gastrulation movements and axis formation to the formation of brain, spinal cord, floor plate, notochord, somites, eyes, ears, heart, blood, pigmentation, vessels, kidney, pharyngeal arches, head skeleton, liver, and gut.

At day 5, formation of swim bladder and overall appearance of the embryos were checked again (at any stage 60–100 embryos were scored). Sixty to hundred fish from heterozygous mutant crosses were grown to adulthood and genotyped to identify the viability of adult homozygous fish. Validated homozygous mutant fish were further crossed together to test for potential fertility phenotypes or putative maternal functions of candidate lncRNAs.

### Antisense RNA synthesis and in situ hybridization

Antisense probes for in situ hybridization were transcribed using the DIG RNA labeling kit (Roche). All RNAs were purified using EZNA Total RNA kits (Omega Biotek). Embryos were fixed in 4% formaldehyde overnight at 4°C (embryos younger than 50% epiboly fixed for 2 days). In situ hybridizations were performed according to standard protocols (*Thisse and Thisse, 2008*). NBT/BCIP/ Alkaline phosphatase-stained embryos were dehydrated in methanol and imaged in benzyl benzoate:benzyl alcohol (BBBA) using a Zeiss Axio Imager.Z1 microscope.

### qRT-PCR

Total RNA was isolated from three individuals or sets of 10–20 embryos per condition using EZNA Total RNA kits (Omega Biotek). cDNA was generated using iScript cDNA Synthesis kit (Bio-Rad). qPCR was conducted using iTaq Universal SYBR Green Supermix (Bio-Rad) on a CFX96 (Bio-Rad). Gene expression levels were calculated relative to a reference gene, *ef1a*. Three technical replicates were used per condition. The qPCR primer sequences are listed in *supplementary file 1*.

### Bright-field imaging

Embryos were anesthetized in Tricaine (Sigma) and mounted in 1% low melting temperature agarose (Sigma) with Tricaine, then imaged using a Zeiss SteREO Discovery.V12 microscope or Zeiss Axio Imager.Z1 microscope. Images were processed in FIJI/ImageJ (*Schindelin et al., 2012*). Brightness, contrast and color balance was applied uniformly to images.

### Sense RNA synthesis and injections

The sequences for the wild-type *squint* mRNA, non-protein coding *squint* transcript (One Adenine base was added after eight in-frame ATG codons, and the 3'UTR sequence kept unchanged) and the *squint$^{a175}$* transcript were synthesized as gBlocks (IDT) containing 5' XhoI cut site and 3' NotI site. Fragments were digested and inserted the pCS2 plasmid. Positive colonies were selected, and sanger sequenced to assure the accuracy of the gene synthesis process. Sequences of the constructs are provided in *supplementary file 1*. mRNA was in vitro transcribed by mMessage mMachine (Ambion) and purified by EZNA Total RNA kits (Omega Biotek). *h2b-gfp* was used as control mRNA. Each injection mix contained 30 ng/ul of *squint* or control mRNA). 1 nl of mRNA mix was injected into the yolk of one-cell stage embryos.

Morpholinos were ordered from Gene Tools and injected based on *Ulitsky et al. (2011)*.

## Acknowledgements

We thank current and former members of the Schier laboratory, particularly Andrea Pauli and Guo-Liang Chew for their helpful suggestions and support during the early phases of this project, Jeffrey Farrell, Nathan Lord and Maxwell Shafer for their critical comments on the manuscript, and the

Harvard zebrafish facility staff for technical support. This work was supported by Leopoldina post-doctoral fellowship LPDS2014-01 to MG and NIH grant R01HD076708 to AFS.

## Additional information

### Funding

| Funder | Grant reference number | Author |
| --- | --- | --- |
| NIH Office of the Director | R01HD076708 | Alexander F Schier |
| Leopoldina | Postdoctorial fellowship LPDS2014-01 | Mehdi Goudarzi |

The funders had no role in study design, data collection and interpretation, or the decision to submit the work for publication.

### Author contributions

Mehdi Goudarzi, Conceptualization, Data curation, Formal analysis, Supervision, Validation, Investigation, Visualization, Methodology, Writing—original draft, Project administration, Writing—review and editing; Kathryn Berg, Lindsey M Pieper, Investigation; Alexander F Schier, Conceptualization, Resources, Supervision, Funding acquisition, Writing—original draft, Project administration, Writing—review and editing

### Author ORCIDs

Mehdi Goudarzi (iD) http://orcid.org/0000-0001-6669-5800
Alexander F Schier (iD) http://orcid.org/0000-0001-7645-5325

### Decision letter and Author response

Decision letter https://doi.org/10.7554/eLife.40815.047
Author response https://doi.org/10.7554/eLife.40815.048

## Additional files

### Supplementary files

• Supplementary file 1. This compressed folder contains three Excel files for the sequences of gRNAs, genotyping and qRT-PCR primers (for lncRNAs and their neighboring genes) and also the annotated sequence files (.ape) for each lncRNA and their deleted segments.
DOI: https://doi.org/10.7554/eLife.40815.014

• Supplementary file 2. This genome-browser-compatible file is in the bed formant, containing the coordinates for all the lncRNAs investigated in this manuscript based on the GRCz11 (GCA_000002035.4).
DOI: https://doi.org/10.7554/eLife.40815.015

• Transparent reporting form
DOI: https://doi.org/10.7554/eLife.40815.016

### Data availability

All data generated or analyzed during this study are included in the manuscript and supporting files. Previously published datasets used in this manuscript: SRP013950, GSE32880, GSE37453, GSE32898, GSE46512

The following previously published datasets were used:

| Author(s) | Year | Dataset title | Dataset URL | Database and Identifier |
| --- | --- | --- | --- | --- |
| Haberle V | 2014 | Two independent transcription initiation codes overlap on vertebrate core promoters. | https://www.ncbi.nlm.nih.gov/bioproject/PRJNA169500 | NCBI Bioproject, SRP013950 |
| Ulitsky I | 2011 | Conserved function of lincRNAs in | https://www.ncbi.nlm. | NCBI Gene |

| | | | | |
|---|---|---|---|---|
| | | vertebrate embryonic development despite rapid sequence evolution. | nih.gov/bioproject/? term=GSE32880 | Expression Omnibus, GSE32880 |
| Ulitsky I | 2012 | Extensive alternative polyadenylation during zebrafish development. | https://www.ncbi.nlm. nih.gov/bioproject/? term=GSE37453 | NCBI Gene Expression Omnibus, GSE37453 |
| Pauli A | 2012 | Systematic identification of long noncoding RNAs expressed during zebrafish embryogenesis. | https://www.ncbi.nlm. nih.gov/bioproject/? term=GSE32898 | NCBI Gene Expression Omnibus, GSE32898 |
| Guo-Liang Chew | 2013 | Ribosome profiling reveals resemblance between long non-coding RNAs and 5' leaders of coding RNAs. | https://www.ncbi.nlm. nih.gov/bioproject/? term=GSE46512 | NCBI Gene Expression Omnibus, GSE46512 |

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
