## [Decision Letter]

Thank you for submitting your article "Long non-coding RNAs are largely dispensable for zebrafish embryogenesis, viability and fertility" for consideration by *eLife*. Your article has been reviewed by three peer reviewers, including Elisabeth M Busch-Nentwich as the Reviewing Editor and Reviewer #1, and the evaluation has been overseen by Didier Stainier as the Senior Editor. The following individual involved in the review of your submission has agreed to reveal their identity: Shawn Burgess (Reviewer #3).

The reviewers have discussed the reviews with one another and the Reviewing Editor has drafted this decision to help you prepare a revised submission.

Summary:

This study describes the generation and morphological characterisation of whole gene or transcription start site deletion mutants in 25 zebrafish lncRNAs. The authors find that the mutants display neither overt developmental phenotypes nor viability or fertility defects. The phenotype of one of the whole gene deletion mutants is due to loss of regulatory elements of neighbouring protein-coding genes which reside within the deleted lncRNA. The authors conclude that, based on these 25 examples, most zebrafish lncRNAs are dispensable for development, viability and fertility.

The experiments are well designed and carefully controlled. The results challenge previous findings using knockdown strategies and will be of high interest.

Essential revisions:

1) The title is not fully supported by the data. "Largely dispensable" implies that some phenotypes have been observed. Also, the extrapolation to all lncRNAs suggested in the title is overreaching based on 25 genes. A less general title, e.g. "Zebrafish embryogenesis, viability and fertility are not overtly affected by loss of embryonically expressed long non-coding RNAs", would be more appropriate. Similarly, in the abstract, the authors state that the lncRNAs are "dispensable for embryogenesis." We suggest softening that statement. There has been a high profile paper that has walked back similar assertions about ultraconserved non-coding sequences (Dickel, 2018). There could easily be (as the authors acknowledge in the conclusions) subtle differences in embryogenesis not detected by eye but that are important to fish in the wild. Saying the fish can survive without the lncRNAs is as bold a statement as you can make with the data presented.

The authors state "Previous large-scale screens have shown that the visual assessment of live embryos and larvae is a powerful and efficient approach to identify mutant phenotypes, ranging from gastrulation movements […] to the formation of brain, […] and gut." This is certainly true, but it is also true that many phenotypes are missed at this level of scrutiny and a sentence admitting that should be included.

2) More detail needs to be provided on the chosen lncRNAs. The selection criteria need to be described more thoroughly. How was evolutionary conservation determined? Conservation of synteny or sequence conservation? What are the thresholds for these criteria (percentage sequence conservation, conserved relative position in which species etc.)? Also, please provide Figure 1 as a table rather than an image and provide the used lncRNA gene models as files that can be uploaded and viewed in a genome browser.

Does expression of the lncRNAs correlate with their neighbours? The authors show expression dynamics across development based on RNA-seq data from their 2012 publication. In the meantime more detailed gene expression datasets have been published, not least single cell data from the same lab. It would be interesting to see whether some of the early expressed lncRNAs can be detected in the single cell data and whether any lineage commitment can be identified.

How do previously described targets (e.g. trans targets for gas5) behave in the mutants?

3) There has been controversy around the squint locus for some time, and while the presented data are fairly convincing, they are not definitive in ruling out squint as a cncRNA family member, as there are still pieces of the gene left both 5' and 3' that could maintain residual function. To maintain the assertion, a complete deletion would be required. Alternatively, it would be acceptable to acknowledge that there may still be an (admittedly small) chance that the remaining gene sequences could have residual non-coding functions.

4) The 500-fold increase of *setdb1a* in the *lnc-setdb1a* mutant is very interesting, but not investigated any further. The authors state that *lnc-setdb1a* mutants are viable and fertile, but it is possible that maternal-zygotic mutants show impaired fertilisation or develop a gametogenesis defect over time. Have the authors looked at fertilisation rates in successive clutches from MZ mutant lnc-setdb1a adults? Is there an effect on setdb1a target gene expression in oocytes?

5) Data on relative expression: Statistical tests should be used to determine whether the difference between wild type and mutant is significant. Further, relative expression levels are based on three biological replicates with three technical replicates each. This means a maximum of nine points underlie each bar in the bar charts. Similarly, N=4 in panel 5E. It would therefore increase data transparency to show the individual data points (and mean and SD) instead of bar charts.

In Figure 3D, Figure 4E, Figure 5C and Figure 6E it is not clear what the expression is relative to. Only Figure legend 5C states "qRT-PCR […] on wild-type and MZ squint embryos", which suggests all measurements are normalised to ef1a levels (as stated in Materials and methods section) and then the wild-type level is set to 1 for each measured gene, but this needs to be clarified in the figure legends.

---

## [Author Response]

Essential revisions:1) The title is not fully supported by the data. "Largely dispensable" implies that some phenotypes have been observed. Also, the extrapolation to all lncRNAs suggested in the title is overreaching based on 25 genes. A less general title, e.g. "Zebrafish embryogenesis, viability and fertility are not overtly affected by loss of embryonically expressed long non-coding RNAs", would be more appropriate. Similarly, in the abstract, the authors state that the lncRNAs are "dispensable for embryogenesis." We suggest softening that statement. There has been a high profile paper that has walked back similar assertions about ultraconserved non-coding sequences (Dickel, 2018). There could easily be (as the authors acknowledge in the conclusions) subtle differences in embryogenesis not detected by eye but that are important to fish in the wild. Saying the fish can survive without the lncRNAs is as bold a statement as you can make with the data presented.

We have changed the Title to:

“Individual long non-coding RNAs have no overt functions in zebrafish embryogenesis, viability and fertility”

We also modified the last sentence of the Abstract to: “LncRNAs might have redundant, subtle, or context-dependent roles, but extrapolation from our results suggests that the majority of individual zebrafish lncRNAs have no overt roles in embryogenesis, viability and fertility”.

The authors state "Previous large-scale screens have shown that the visual assessment of live embryos and larvae is a powerful and efficient approach to identify mutant phenotypes, ranging from gastrulation movements […] to the formation of brain, […] and gut." This is certainly true, but it is also true that many phenotypes are missed at this level of scrutiny and a sentence admitting that should be included.

We added to the Discussion section: “Our phenotypic screen does not exclude more subtle phenotypes; e.g. in behavior or brain activity”.

2) More detail needs to be provided on the chosen lncRNAs. The selection criteria need to be described more thoroughly. How was evolutionary conservation determined? Conservation of synteny or sequence conservation? What are the thresholds for these criteria (percentage sequence conservation, conserved relative position in which species etc.)?

In the current manuscript we have state that:

“For our knockout study we selected 24 bona fide lncRNAs based on syntenic and sequence conservation, expression dynamics and proximity to developmental regulatory genes (see Table 1). These criteria were chosen to increase the likelihood of functional requirement.”

We have revised the manuscript as follows:

“For our mutant analysis we selected 24 bona fide lncRNAs based on synteny (conserved relative position on at least one other vertebrate genome), sequence conservation, expression dynamics (expression levels, onset and pattern) and proximity to developmental regulatory genes (see Table 1). These criteria were chosen to increase the likelihood of potential functional requirements of the selected lncRNAs.”

We have also extended the section “selection criteria” in Table 1 to better represent the main selection criteria for each lncRNA.

Also, please provide Figure 1 as a table rather than an image and provide the used lncRNA gene models as files that can be uploaded and viewed in a genome browser.

We replaced Figure 1 with Table 1 and provided a genome-browser-compatible file for the coordinates of the studied lncRNAs (Supplementary file 2).

Does expression of the lncRNAs correlate with their neighbours?

We did not systematically investigate this question, but we have provided the expression dynamics of investigated lncRNAs and their immediate neighbors in a 200kb window. In our selection, we did not find unifying themes for co-expression patterns of lncRNAs and their immediate neighboring genes.

The authors show expression dynamics across development based on RNA-seq data from their 2012 publication. In the meantime more detailed gene expression datasets have been published, not least single cell data from the same lab. It would be interesting to see whether some of the early expressed lncRNAs can be detected in the single cell data and whether any lineage commitment can be identified.

Based on this suggestion, we tested all of our lncRNAs with assigned transcript ID, in the single cell data from Farrell et al., 2018. We observe clear trajectory restriction only for squint and a potential partial enrichment might exist for lnc-3852. Expression distribution and trajectory trees for these lncRNAs are provided in Author response images 1-10. No data could be retrieved for four lncRNAs (transcript IDs Lnc-phox2bb, lnc-2646, lnc4468, lnc1666). Although not very informative, we can add this analysis to the paper if the reviewers wish.

**Author Response image 1. respfig1:** 

**Author Response image 2. respfig2:** 

**Author Response image 3. respfig3:** 

**Author Response image 4. respfig4:** Clear enrichment in the margin and prechordal plate cells.

**Author Response image 5. respfig5:** Some partial enrichment in the Neural Plate Border cells and Somites.

**Author Response image 6. respfig6:** 

**Author Response image 7. respfig7:** 

**Author Response image 8. respfig8:** 

**Author Response image 9. respfig9:** 

**Author Response image 10. respfig10:** 

**Author Response image 11. respfig11:** 

**Author Response image 12. respfig12:** 

**Author Response image 13. respfig13:** 

**Author Response image 14. respfig14:** 

**Author Response image 15. respfig15:** 

**Author Response image 16. respfig16:** 

**Author Response image 17. respfig17:** 

How do previously described targets (e.g. trans targets for gas5) behave in the mutants?

Previous studies have shown that gas5 lncRNA can act in trans to affect pten expression (ptena and ptenb in zebrafish) by sequestering specific microRNAs including miR-103, miR-222 and miR-21. Additionally, gas5 transcript can mimic Glucocorticoid Response Element and act as a decoy factor for the Glucocorticoid Receptor (nr3c1) mediated transcription. We analyzed the expression level changes of these genes in MZgas5^a173^ embryos (at 1-dpf) relative to WT by qRT-PCR using previously published primer pairs for these genes in zebrafish. T-test analysis revealed significant upregulation for ptena in MZgas5^a173^ mutants.

**Author Response image 18. respfig18:** 

3) There has been controversy around the squint locus for some time, and while the presented data are fairly convincing, they are not definitive in ruling out squint as a cncRNA family member, as there are still pieces of the gene left both 5' and 3' that could maintain residual function. To maintain the assertion, a complete deletion would be required. Alternatively, it would be acceptable to acknowledge that there may still be an (admittedly small) chance that the remaining gene sequences could have residual non-coding functions.

The previously published claims were related to the function of specific conserved element in the 3’UTR of squint. Our deletion mutant is sufficient to rule out the proposed dorsalizing function for the 3’UTR of maternal squint. In our mutant allele, 525 basepairs comprising of 5’UTR (178bp), first exon (280bp) and part of the second exon (67bp) remain in the genome. The above-mentioned sequences encode part of the squint prodomain and might still provide some unclaimed non-coding function.

We revised text as follows:

“To investigate the non-coding roles of squint mRNA we generated a deletion allele (squint^a175^) that lacked most of the protein coding region and the 3’UTR, including the Dorsal Localization Element (DLE) implicated in maternal squint RNA localization^52^ (Figure 4A). In this allele 525bp (178bp 5’UTR, 280bp first exon and 67bp of second exon) out of the 1592bp-long mature transcript remain in the genome (Figure 4A).”

“These results indicate that squint 3’UTR does not have the previously proposed noncoding functions and that the squint transcript may not be a member of the cncRNA family.”

4) The 500-fold increase of setdb1a in the lnc-setdb1a mutant is very interesting, but not investigated any further. The authors state that lnc-setdb1a mutants are viable and fertile, but it is possible that maternal-zygotic mutants show impaired fertilisation or develop a gametogenesis defect over time. Have the authors looked at fertilisation rates in successive clutches from MZ mutant lnc-setdb1a adults? Is there an effect on setdb1a target gene expression in oocytes?

We did not systematically quantify the clutch size and fertilization rates. We scored four independent crosses of Wild type -AB (14-month old) and MZlnc-setd1ba^a176^ fish (17-month old), that resulted in clutches which did not show significant differences by multiple t-test analysis in any of the assayed categories (Unfertilized, Dead at 24hpf and Total number).

**Author Response image 19. respfig19:** 

5) Data on relative expression: Statistical tests should be used to determine whether the difference between wild type and mutant is significant. Further, relative expression levels are based on three biological replicates with three technical replicates each. This means a maximum of nine points underlie each bar in the bar charts. Similarly, N=4 in panel 5E. It would therefore increase data transparency to show the individual data points (and mean and SD) instead of bar charts. In Figure 3D, Figure 4E, Figure 5C and Figure 6E it is not clear what the expression is relative to. Only Figure legend 5C states "qRT-PCR […]) on wild-type and MZ squint embryos", which suggests all measurements are normalised to ef1a levels (as stated in Materials and methods section) and then the wild-type level is set to 1 for each measured gene, but this needs to be clarified in the figure legends.

All the bar-graphs in the main figures are changed to represent individual data points, and appropriate statistical analysis is performed to define the significance of observed differences.